# Quantitative analysis of infection dynamics of foot-and-mouth disease virus strain O/CATHAY in pigs and cattle

**Tatsuya Nishi[1], Kazuki Morioka[1], Rie Kawaguchi[1], Manabu Yamada[2], Mitsutaka Ikezawa[2], Katsuhiko Fukai [1]***

**1** Exotic Disease Research Station, National Institute of Animal Health, National Agriculture and Food Research Organization, Kodaira, Tokyo, Japan, **2** Division of Pathology and Pathophysiology, National Institute of Animal Health, National Agriculture and Food Research Organization, Tsukuba, Ibaraki, Japan

* fukai@affrc.go.jp

**Data Availability Statement:** All relevant data are within the manuscript, tables and figures.

**Funding:** Ministry of Agriculture, Forestry and Fisheries, Japan.

## Abstract

Foot-and-mouth disease virus (FMDV) serotype O, topotype CATHAY is a known porcinophilic virus that has caused devastating damage to the pig industry. However, the minimum infectious dose via a natural infection route in pigs, the infection dynamics in cattle, and risk of viral transmission from infected cattle to pigs have not been quantitatively analyzed. The FMDV strain O/HKN/1/2015 was serially diluted and inoculated into pigs via an intraoral route to determine the infectious dose. We found that a $10^{4.0}$ tissue culture infectious dose ($TCID_{50}$) of the virus was insufficient, but $10^{5.5}$ $TCID_{50}$ was sufficient to infect pigs via the oral route. While cows inoculated with the strain showed increased temperature in their feet, typical clinical signs including vesicular development were not observed. The cows showed short-term and low levels of viremia and virus excretion only before the detection of virus neutralizing antibodies. FMDV genes were not detected in esophageal-pharyngeal fluid from cows after 14 days post inoculation. No genetic insertions that could be associated with host adaptation were observed in viruses isolated from infected cows. These findings indicate that cows infected with FMDV of O/CATHAY have a low risk of viral transmission or persistence. Information on the dynamics of virus infection is essential for ensuring the rapid and accurate diagnosis of this disease, and its surveillance.

## Introduction

Foot-and-mouth disease virus (FMDV) infects a wide range of cloven-hoofed animals and causes severe economic damage to livestock industries [1,2]. FMDV-infected animals display typical lesions comprising vesicles around the mouth, snout, mammary glands and feet. Control of FMD is impeded by its transmissibility across different or multiple species and the existence of a persistent infection in ruminant species [3,4]. However, while pigs shed high concentrations of virus in their breath but are relatively resistant to airborne infection, cattle and other ruminants are highly sensitive to airborne infection [5]. Persistent FMDV infection, defined by detection of infectious FMDV in esophageal-pharyngeal (OP) fluid more than 15

**Competing interests:** The authors have declared that no competing interests exist.

or 21 days post-infection (dpi) in vaccinated or naive cattle, respectively, occurs in more than 50% of cases in both groups, regardless of the occurrence of clinical disease [6–8].

In 1997, a FMDV confirmed in Taiwan showed atypical pathogenicity with high morbidity and mortality in swine but no effect on cattle [9]. This devastating outbreak led to the culling of over 4 million pigs and severe economic losses. The causative agent was confirmed to be a distinct topotype of serotype O (O/CATHAY), identified for the first time in 1970 in Hong Kong SAR and China, characterized by a deletion within the 3A and pseudoknot regions of its genome and low infectivity for cattle [10–13]. Since the catastrophic outbreak in Taiwan, sporadic outbreaks caused by O/CATHAY strains have been reported in Taiwan, Hong Kong SAR and China, and circulating in several Southeast Asian countries together with FMDVs of O/SEA/Mya-98, O/ME-SA/PanAsia and O/ME-SA/Ind-2001 [14]. According to quarterly reports from the World Reference Laboratory for Foot-and-Mouth Disease (https://www.wrlfmd.org/ref-lab-reports), low antigenic matching between the O/CATHAY strain and current vaccine strains make the disease difficult to control. Therefore, current strategies to eradicate FMDV of this topotype rely on the rapid detection of infected animals and control measures including movement restriction and culling of animals suspected of infection.

Previous studies investigated the infection dynamics of O/TAW/1997 isolated from the outbreak in Taiwan by inoculating pigs through the intradermal route via the heel bulb [15,16]. While these studies identified the amount of virus excreted, transmission to contact animals, and pathology and viral distribution of infected pigs, they did not determine the minimum infectious dose required to cause disease in pigs via a natural infection route. In addition, to our knowledge, the dynamics of the porcinophilic virus in cows, possibility of adaptation or persistence, and the time-course of antibody response in them which are required for appropriate diagnosis and surveillance have not been reported.

Here, to better understand the diagnosis and control measures for rapid eradication of FMD outbreaks caused by an FMDV O/CATHAY strain, we investigate the dynamics of infection, including manifestation of clinical signs, virus excretion and antibody response, in pigs and cows experimentally infected with the porcinophilic strain of FMDV O/HKN/1/2015.

## Materials and methods

### Experimental infection of pigs and cattle

Two pigs each were orally inoculated with 1ml of a $10^{4.0}$ (pigs no. 1 and 2), $10^{5.5}$ (no. 3 and 4) and $10^{7.0}$ (no. 5 and 6) tissue culture infectious dose ($TCID_{50}$)/ml of FMDV O/HKN/1/2015 titrated using LFPK-$\alpha v\beta 6$ cells [17,18]. Each pair of pigs was housed in one cubicle. Clinical signs were observed daily and sera, oral and nasal swab samples were collected for approximately 14 days as described previously [4]. Two six-year-old Japanese Black cows were sedated with xylazine intramuscular injection (0.05 mg/kg) and were subepidermo-lingually inoculated with 1 ml of $10^{7.0}$ $TCID_{50}$/ml of FMDV O/HKN/1/2015 [19]. The animals were housed separately in two cubicles for 24 days. In addition to sera, oral and nasal swab samples, OP fluid was collected using a probang cup from the cows at 0, 10, 14, 17, 21, and 24 dpi. Clinical signs were scored as follows: lesion in main hoof, 1 point per foot; lesions in accessory digit, 1 point per foot; lesions in the snout, 1 point; fever (40°C or more), anorexia or dullness, 1 point. The maximum score per animal was 10.

All animal procedure was approved prior to the initiation of this study by the Animal Care and Use Committee of the National Institute of Animal Health (NIAH), which functions to ensure ethical and humane treatment and animal welfare of experimental animals (Authorization Numbers: 18–065 and 19–094, approved 20 December, 2018 and 28 February, 2020, respectively). All experimental infections using live viruses were performed in cubicles of

approximately 14 m$^2$ kept at 25°C with 10 to 15 air changes per hour in a high-containment facility at the NIAH. During the experiments, rectal temperatures of animals were taken and their behavior and physiology were observed by veterinarians daily. All pigs and cattle survived until the end of the experimental period, and after that the animals were euthanized by an injection of ketamine or sodium pentobarbital and subjected to necropsy.

## Virus titration

LFPK-$\alpha v\beta 6$ cells were used for virus titration. Virus titration was performed as follows: serial 10-fold dilutions of stock viruses were prepared in tubes, and each dilution was inoculated into the same volume of a suspension of LFPK-$\alpha v\beta 6$ cells in 10 wells of 96-well plates. The cultures were incubated for 72 hr at 37°C in 5% $CO_2$ and monitored for the appearance of any cytopathic effects. Virus titers of $TCID_{50}$ were calculated using the Reed-Muench method.

## RNA extraction, real-time reverse transcription-PCR (RT-PCR), and sequencing

Viral RNA was extracted from clinical samples using the High Pure Viral RNA Kit (Roche Diagnostics, Tokyo, Japan). FMDV-specific genes were detected using the TaqMan Fast Virus 1-Step Master Mix (Life Technologies) with 900 nM of primer sets and 250 nM of probe targeting the 3D region [20]. In the present study, samples with a Ct value below 37 were defined as positive. The full length of the L-fragment gene of approximately 7.7 kb was amplified by RT-PCR using SuperScript IV One-Step RT-PCR System (Life Technologies) and two primer sets: set 1 consisted of a 5'-NT F primer (5'-CCGTCGTTCCCGACGTTAAAGGG-3') and 2B331R primer (5'-GGCACGTGAAAGAGACTGGAGAG-3') and set 2 consisted of a 2B217F primer (5'-ATGGCCGCTGTAGCAGCACGGTC-3') and 3'-NT R primer (5'-CAATTGGCAGAAAGA CTCTGAGGCG-3'). Their nucleotide sequences were analyzed using the Ion PGM system (Life Technologies).

## Antibody detection from sera

Virus neutralization test (VNT) was performed using BHK-21 cells as described in the terrestrial manual, 2019 [21]. O/HKN/1/2015 was used as an antigen in the VNT to determine the antibody response to the virus in infected animals. FMDV-specific antibodies were detected using the PrioCHECK FMDV Type O Antibody ELISA Kit (Applied Biosystems, California, USA) and a solid phase competitive ELISA kit (SPCE) (IZLER, Brescia, Italy).

## Results

### Dose-dependent experimental infection of pigs with O/HKN/1/2015

Pigs no. 1 and 2 showed no clinical signs and their clinical samples were negative for virus titers. Their sera were negative for antibodies against FMDV, indicating that they had not been infected with the virus (Tables 1–3). In contrast, pigs no. 3 and 4 showed pyrexia and vesicular lesions on their feet from 3 or 4 dpi, and lesions in the snout and marked anorexia from 5 dpi (Table 1). Both pigs had a total clinical score of 10 (Fig 1). Additionally, viremia was confirmed 3–4 or 2–4 dpi, and virus excretion ranging from $10^{4.0}$ to $10^{8.5}$ $TCID_{50}$/ml (titrated in LFPK-$\alpha v\beta 6$ cells) from oral and nasal discharge was confirmed 2–7 or 1–8 dpi. Antibodies were detected from 5 dpi using VNT, and from 7 dpi using the PrioCHECK antibody ELISA Kit (Table 3).

Pigs no. 5 and 6, which were inoculated with $10^{1.5}$-fold higher titers of the virus than pigs no. 3 and 4, showed signs of infection approximately one day earlier (Fig 1, Tables 1–3).

**Table 1. Post-inoculation day on which clinical signs were initially observed in pigs and cows infected with FMDV O/HKN/1/2015.**

| Clinical sign | | | Inoculated TCID$_{50}$ titer | | | | | | |
|---|---|---|---|---|---|---|---|---|---|
| | | $10^4$ | | $10^{5.5}$ | | $10^7$ | | $10^7$ | |
| | | Pig no.1 | Pig no.2 | Pig no.3 | Pig no.4 | Pig no.5 | Pig no.6 | Cow no.1 | Cow no.2 |
| Pyrexia | | – | – | 4 | 3 | 3 | 3 | – | – |
| Anorexia | | – | – | 5 | 5 | 3 | 3 | 2 | 2 |
| Vesicular development | | | | | | | | | |
| Snout | | – | – | 5 | 5 | 4 | – | – | – |
| Forelimb | Right | – | – | 4 | 3 | 2 | 3 | – | – |
| | Left | – | – | 4 | 3 | 2 | 3 | – | – |
| Hindlimb | Right | – | – | 4 | 3 | 2 | 2 | – | – |
| | Left | – | – | 4 | 3 | 2 | 3 | – | – |

–, Not detected.

Namely, pyrexia and vesicular lesions were observed from 2 or 3 dpi, viremia from 2–3 or 2–4 dpi, virus excretion ranging from $10^{3.5}$ to $10^{8.5}$ TCID$_{50}$/ml from 1–7 or 10 dpi, antibody detection from 4 dpi using VNT and from 5 or 6 dpi using the PrioCHECK antibody ELISA Kit. Among pigs no. 3–6, sera from pig no. 6 only were positive using SPCE between 6 and 10 dpi (Table 3).

## Virus challenge test to cattle with O/HKN/1/2015

Both inoculated cows showed signs of anorexia from 2 dpi, but no vesicular lesions (Table 1). The total clinical score was 1. However, infrared thermography (IRT) imaging using an FLIR C2 camera (FLIRSystems, Oregon, USA) demonstrated that the temperature of the hind limbs of cow no. 1 and the four limbs of cow no. 2 was markedly high between 3 and 5 dpi (Fig 2). While viremia was confirmed between 1 and 3 dpi in both cows, virus excretion was limited to only oral discharge in cow no. 1 at 2 dpi, although FMDV genes were detected in oral and nasal discharge 1–5 or 3–5 dpi, and OP fluid from cow no. 2 at 10 dpi (Fig 1, Table 2). Antibodies were detected from 4 or 5 dpi using VNT, 4 or 7 dpi using the PrioCHECK ELISA kit, and after 10 dpi using SPCE (Table 3).

The full length of the L-fragment gene of O/HKN/1/2015 isolated from sera from cows no. 1 and 2, collected at 1 and 2 dpi, respectively, aligned perfectly with that from the virus stock. This indicates that there were no changes to deletions of 10 amino acid within the 3A (Table 4) or 43 nucleotides within the pseudoknot regions (S1 File).

## Discussion

Risk analysis on viral infection in animals and the development of control measures requires information on viral load and excretion dynamics in infected animals, as well as the infectious dose required to cause disease. FMD can spread via a variety of routes, including intranasal, intraoral, and intradermal routes [1,2]. Previous studies have demonstrated that, in contrast to cows, pigs are more refractory to aerogenous infection but more susceptible to infection via the gastrointestinal route, and excrete greater quantities of virus [5,15]. A previous study conducted in our institute using the same method as in the present study revealed that pigs were more susceptible to infection via the intraoral than intranasal route [22]. Therefore, in this study, we performed inoculation under conditions that are most likely to cause infection in pigs via the natural infection route. Pigs inoculated with $10^{5.5}$ and $10^{7.0}$ TCID$_{50}$ O/HKN/1/2015 showed typical clinical signs of FMD and excreted $10^{3.5}$ to $10^{8.5}$ TCID$_{50}$/ml of virus (Fig

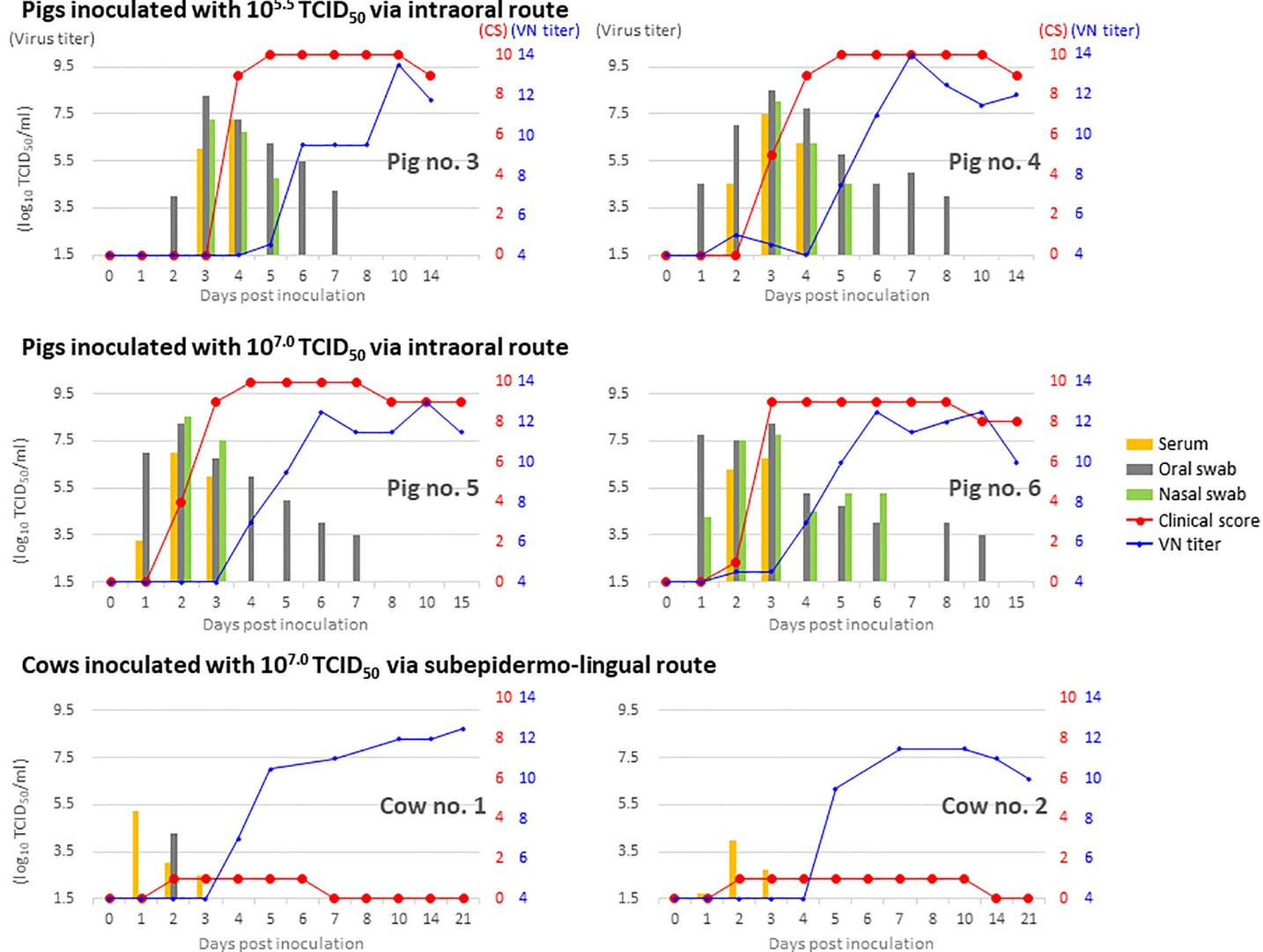

**Fig 1. Dynamics of FMDV infection with the O/HKN/1/2015 strain in pigs and cows.** Two pigs each were inoculated with $10^{5.5}$ and $10^{7.0}$ tissue culture infectious dose ($TCID_{50}$) of the virus via the intraoral route. Cows no. 1 and 2 were inoculated with $10^{7.0}$ $TCID_{50}$ via the subepidermo-lingual route. The x-axis shows the number of days post infection (dpi). Viral titers (as $log_{10}$ $TCID_{50}$/ml) in sera and oral and nasal swab samples are shown yellow, grey, green, respectively on the y-axis on the left, and development of clinical signs (CS, score from 0 to 10; red) and virus neutralization (VN) titers (as $log_2$; blue) are shown on the y-axis on the right. The two pigs (no. 1 and 2) inoculated with $10^{4.0}$ $TCID_{50}$ of O/HKN/1/2015 had a clinical score of zero and were negative for virus and VN titers.

1). In contrast, two pigs (no. 1 and 2) inoculated with $10^{4.0}$ $TCID_{50}$ of O/HKN/1/2015 did not show signs of infection. These data indicate that more than $10^{4.0}$ $TCID_{50}$ of the virus is required to infect pigs via the oral route. This report demonstrates the infectious dose of a recent isolate of O/CATHAY in pigs and proves that infected pigs excrete sufficient levels of virus to infect other pigs.

A number of previous reports have demonstrated that cows infected with typical FMDV excrete high amounts of virus, although less than pigs, making them a high risk animal for FMDV transmission [15,23]. A high rate of persistent infection in ruminants can also inhibit complete eradication of FMDV [3]. In the previous study, we subepidermo-lingually inoculated the tongues of Japanese Black cows with O/HKN/1/2015. This is considered the most effective route for infecting cattle because subepidermo-lingual inoculation is used for FMDV

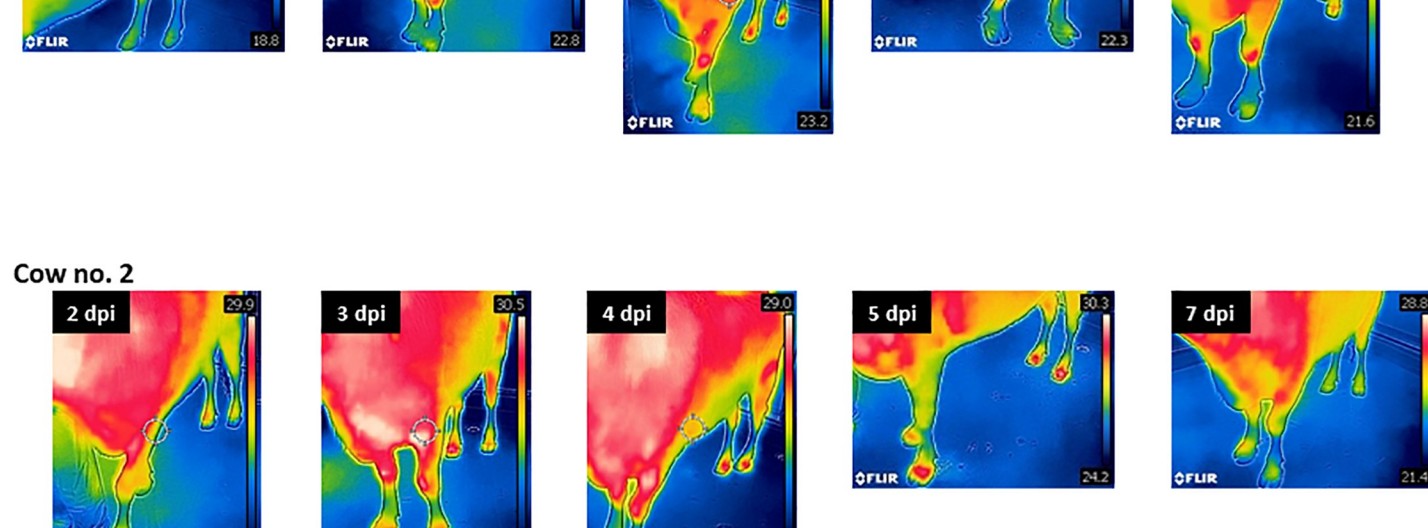

**Fig 2. Infrared images of cows at each day post inoculation.** Lower temperatures are indicated in blue–green and higher temperatures in orange–red. Temperatures of hoofs of the hind limbs of cow no. 1 and the four limbs of cow no. 2 were markedly high between 3 and 5 days post infection (dpi).

challenge, and Japanese Black cattle are thought to be more susceptible to FMDV than Holstein cattle, based on previous reports and findings in the field [24]. Although this experimental approach is highly artificial, findings could indicate the dynamics of the porcinophilic virus including capability of propagation, adaptation or persistence in cows, and the time-course of antibody response for appropriate diagnosis. In the present study, inoculated Japanese Black cows only showed clinical signs of anorexia. This finding is consistent with that of a previous study describing the slight clinical signs of cattle intradermally inoculated on the tongues with O/TAW/1997 [9]. Quantities of virus excreted from infected cows, which were reported for the first time in this study, were less than $10^{4.3}$ TCID$_{50}$/ml (Fig 1). This can be speculated to be insufficient to infect pigs considering the infectious dose confirmed in this study, although further experimental contact transmission studies are required to conclude their incapability of infecting other animals. Nevertheless, virus neutralizing antibodies were detected during almost the same period as that in pigs. Although FMDV genes were detected in OP fluid from one cow at 10 dpi, they were no longer detectable after 14 dpi (Table 2). No genetic changes that could be associated with host adaptation were present in virus isolated from the sera of infected cows (Table 4). These data indicate that O/HKN/1/2015 could not effectively propagate in cattle and was rapidly neutralized without causing disease. Therefore, cows infected with FMDV of O/CATHAY have a low risk of viral transmission or persistence. It is expected to be one major reason for the smaller number of outbreaks caused by this than other topotypes. Surprisingly, however, sporadic outbreaks continue to be reported in several Southeast Asian countries; where and how viruses of this topotype are maintained or spread remain unclear [14,25]. Because current vaccines have low potency against the O/CATHAY strain, in the event of an outbreak, strategies to eradicate FMDV of this topotype rely on movement

**Table 2. Time-course of detection of FMDV-specific genes in samples from pigs and cows inoculated with O/HKN/1/2015.**

| Inoculated TCID$_{50}$ | Animal no. | Sample | Detection of FMDV-specific genes at each day post infection | | | | | | | | | | | | |
|---|---|---|---|---|---|---|---|---|---|---|---|---|---|---|---|
| | | | 0 | 1 | 2 | 3 | 4 | 5 | 6 | 7 | 8 | 10 | 14[a] | 21 | 24 |
| 10$^4$ | Pig no. 1 | Serum | – | – | – | – | – | – | – | – | – | – | – | ns | ns |
| | | Oral swab | – | – | – | – | – | – | – | – | – | – | – | ns | ns |
| | | Nasal swab | – | – | – | – | – | – | – | – | – | – | – | ns | ns |
| | Pig no. 2 | Serum | – | – | – | – | – | – | – | – | – | – | – | ns | ns |
| | | Oral swab | – | – | – | – | – | – | – | – | – | – | – | ns | ns |
| | | Nasal swab | – | – | – | – | – | – | – | – | – | – | – | ns | ns |
| 10$^{5.5}$ | Pig no. 3 | Serum | – | – | + | + | + | + | – | – | – | – | – | ns | ns |
| | | Oral swab | – | – | + | + | + | + | + | + | + | + | + | ns | ns |
| | | Nasal swab | – | – | – | + | + | + | + | + | + | + | – | ns | ns |
| | Pig no. 4 | Serum | – | – | – | + | + | + | – | – | – | – | – | ns | ns |
| | | Oral swab | – | + | + | + | + | + | + | + | + | + | + | ns | ns |
| | | Nasal swab | – | – | – | + | + | + | + | + | + | + | – | ns | ns |
| 10$^7$ | Pig no. 5 | Serum | – | – | + | + | – | – | – | – | – | – | – | ns | ns |
| | | Oral swab | – | + | + | + | + | + | + | + | – | + | – | ns | ns |
| | | Nasal swab | – | – | + | + | + | + | + | + | – | – | – | ns | ns |
| | Pig no. 6 | Serum | – | – | + | + | + | – | – | – | – | – | – | ns | ns |
| | | Oral swab | – | + | + | + | + | + | + | + | + | + | – | ns | ns |
| | | Nasal swab | – | + | + | + | + | + | + | + | + | + | – | ns | ns |
| 10$^7$ | Cow no. 1 | Serum | – | + | + | + | – | – | ns | – | ns | – | – | – | – |
| | | Oral swab | – | + | + | + | + | + | ns | – | ns | – | – | – | – |
| | | Nasal swab | – | – | + | – | – | – | ns | – | ns | – | – | – | – |
| | | OP fluid | – | ns | ns | ns | ns | ns | ns | ns | ns | – | – | – | – |
| | Cow no. 2 | Serum | – | – | + | + | – | – | ns | – | ns | – | – | – | – |
| | | Oral swab | – | – | – | + | + | + | ns | – | ns | – | – | – | – |
| | | Nasal swab | – | – | – | + | – | – | ns | – | ns | – | – | – | – |
| | | OP fluid | – | ns | ns | ns | ns | ns | ns | ns | ns | + | – | – | – |

OP, oesophageal-pharyngeal; ns, not sampled.

[a] Samples were collected at 14 dpi (pigs no. 1–4 and cows no. 1–2) and 15 dpi (pigs no. 5–6).

**Table 3. Time-course of detection of FMDV-specific antibodies by ELISA in the sera of pigs and cows inoculated with O/HKN/1/2015.**

| Inoculated TCID$_{50}$ | Animal no. | Detection of FMDV-specific ELISA antibodies at each day post infection | | | | | | | | | | | | |
|---|---|---|---|---|---|---|---|---|---|---|---|---|---|---|
| | | 0 | 1 | 2 | 3 | 4 | 5 | 6 | 7 | 8 | 10 | 14[a] | 21 | 24 |
| 10$^4$ | Pig no. 1 | –/– | –/– | –/– | –/– | –/– | –/– | –/– | –/– | –/– | –/– | –/– | ns | ns |
| | Pig no. 2 | –/– | –/– | –/– | –/– | –/– | –/– | –/– | –/– | –/– | –/– | –/– | ns | ns |
| 10$^{5.5}$ | Pig no. 3 | –/– | –/– | –/– | –/– | –/– | –/– | –/– | +/– | +/– | +/– | +/– | ns | ns |
| | Pig no. 4 | –/– | –/– | –/– | –/– | –/– | –/– | –/– | +/– | +/– | –/– | –/– | ns | ns |
| 10$^7$ | Pig no. 5 | –/– | –/– | –/– | –/– | –/– | –/– | +/– | +/– | +/– | +/– | +/– | ns | ns |
| | Pig no. 6 | –/– | –/– | –/– | –/– | –/– | +/– | +/+ | +/+ | +/+ | +/+ | –/– | ns | ns |
| 10$^7$ | Cow no. 1 | –/– | –/– | –/– | –/– | +/– | +/– | ns | +/– | ns | +/– | +/– | +/– | +/– |
| | Cow no. 2 | –/– | –/– | –/– | –/– | –/– | –/– | ns | +/– | ns | +/+ | +/– | +/– | +/– |

Antibodies against FMDV were detected using the PrioCHECK FMDV Type O Antibody ELISA Kit (left) and SPCE (right).

ns, not sampled.

[a] Samples were collected at 14 dpi (pigs no. 1–4 and cows no. 1–2) and 15 dpi (pigs no. 5–6).

**Table 4. Alignment of predicted amino acid sequences corresponding to codons 80 to 115 of the 3A coding region.**

| | Amino acid at the indicated position in 3A[a] | | | | | | | | | | | | | | | | | | | | | | | | | | | | | | | | | | | |
|---|---|---|---|---|---|---|---|---|---|---|---|---|---|---|---|---|---|---|---|---|---|---|---|---|---|---|---|---|---|---|---|---|---|---|---|---|
| **Virus** | **80** | | | | | | | | | | **90** | | | | | | | | | | **100** | | | | | | | | | | **110** | | | | | | |
| O/HKN/1/2015 | | | | | | | | | | | | | | | | | | | | | | | | | | | | | | | | | | | | | |
| Inoculum | R | K | R | R | Q | S | V | D | D | S | L | D | S | – | – | – | – | – | – | – | – | – | – | D | I | T | L | G | D | A | E | K | N | P | L | E |
| Cow no. 1 (1 dpi) | · | · | · | · | · | · | · | · | · | · | · | · | · | – | – | – | – | – | – | – | – | – | – | · | · | · | · | · | · | · | · | · | · | · | · | · |
| Cow no. 2 (2 dpi) | · | · | · | · | · | · | · | · | · | · | · | · | · | – | – | – | – | – | – | – | – | – | – | · | · | · | · | · | · | · | · | · | · | · | · | · |
| O/JPN/2010-290/1E | · | · | · | Q | · | M | · | · | · | A | V | N | E | Y | I | E | K | A | N | I | T | T | D | · | K | · | · | D | E | · | · | · | · | · | · | · |

Dot (·), identity amino acid with O/HKN/1/2015; Dash (−), amino acid deletion.

[a] Amino acid numbers were annotated with the 3A of O/JPN/2010-290/1E (accession number: LC036265). The sequences of O/HKN/1/2015 (LC595604) of virus stock and sera from cows no. 1 and 2 were indicated.

restriction and culling of animals suspected of infection. Given that the pig industry is one of the most important sectors of agriculture in Asia, such strategies can lead to severe economic losses. Further statistical surveillance is needed for this porcinophillic FMDV to strategize appropriate risk management and to reduce the possibility of virus introduction.

Information on the dynamics of virus infection is essential for ensuring rapid and accurate diagnosis and surveillance of disease, and facilitates appropriate sample collection at each phase of the disease for laboratory analysis. As with typical FMD, we detected maximal concentrations of FMDV in sera and oral and nasal discharge in the early stage of clinical disease, one or two days before the peak clinical score (Fig 1). Diagnosis of FMD in cows in the early stage of disease is difficult because of the limited clinical signs and virus excretion during this period. IRT may be helpful for identifying potential FMDV-infected cows for further sampling [26] (Fig 2). Antibodies were detected in infected cows and pigs using VNT and a commercial ELISA kit, while SPCE showed limited sensitivity, possibly because of antigenic mismatch (Table 3). This information is critically important for the application of serological surveillance for diagnosis and in substantiating freedom from infection in outbreaks caused by FMDV of the CATHAY topotype.

In conclusion, this study quantitatively analyzed infection in pigs and cows experimentally inoculated with a strain of FMDV O/CATHAY. These findings are expected to aid the development of appropriate diagnosis, surveillance, and control measures for rapid eradication of FMD outbreaks.

## Supporting information

**S1 File. The nucleotide sequences of O/HKN/1/2015 obtained from sera from cows no. 1 and 2.**
(TXT)

## Acknowledgments

We would like to acknowledge the Pirbright Institute (UK) for the supply of the FMDV strain. We are grateful to Dr. Satoko Kawaji, Ms. Reiko Nagata, and Dr. Ken-ichiro Kameyama for their valuable support. We also thank Tomoko Kato, Nobuko Saito, and Sachiko Tanamura for their technical assistance. We would also like to thank Mr. Kenichi Ishii, Mr. Masayuki Kanda, Mr. Yu-ki Takahashi, and Mr. Tatsuo Nakamura for their skilled handling of animals at the NIAH.

## Author Contributions

**Conceptualization:** Tatsuya Nishi, Kazuki Morioka, Katsuhiko Fukai.

**Data curation:** Tatsuya Nishi.

**Investigation:** Tatsuya Nishi, Kazuki Morioka, Rie Kawaguchi, Manabu Yamada, Mitsutaka Ikezawa, Katsuhiko Fukai.

**Writing – original draft:** Tatsuya Nishi.

**Writing – review & editing:** Kazuki Morioka, Rie Kawaguchi, Manabu Yamada, Katsuhiko Fukai.

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
