## [Decision Letter · Decision Letter 0]

13 Nov 2020

PONE-D-20-32103

Quantitative analysis of infection dynamics of foot-and-mouth disease virus strain O/CATHAY in pigs and cattle

PLOS ONE

Dear Dr. Fukai,

Thank you for submitting your manuscript to PLOS ONE. After careful consideration, we feel that it has merit but does not fully meet PLOS ONE’s publication criteria as it currently stands. Therefore, we invite you to submit a revised version of the manuscript that addresses the points raised during the review process.

We look forward to receiving your revised manuscript.

Kind regards,

Douglas Gladue, Ph.D

Academic Editor

PLOS ONE

Additional Editor Comments (if provided):

The second reviewer never submitted any comments, however I agree with the first reviewer. Please respond to all of the comments in a point by point response when resubmitting.

Journal Requirements:

2) We note that you have included the phrase “data not shown” in your manuscript. Unfortunately, this does not meet our data sharing requirements. PLOS does not permit references to inaccessible data. We require that authors provide all relevant data within the paper, Supporting Information files, or in an acceptable, public repository. Please add a citation to support this phrase or upload the data that corresponds with these findings to a stable repository (such as Figshare or Dryad) and provide and URLs, DOIs, or accession numbers that may be used to access these data. Or, if the data are not a core part of the research being presented in your study, we ask that you remove the phrase that refers to these data.

3) In your Methods section, please provide additional information on the animal research and ensure you have included details on whether animals died before the end of virus infection experiment; if so, specify the number of animals found dead and the reasons for the death.

Reviewers' comments:

Reviewer's Responses to Questions

**Comments to the Author**

1. Is the manuscript technically sound, and do the data support the conclusions?

Reviewer #1: Partly

2. Has the statistical analysis been performed appropriately and rigorously? 

Reviewer #1: N/A

3. Have the authors made all data underlying the findings in their manuscript fully available?

Reviewer #1: Yes

4. Is the manuscript presented in an intelligible fashion and written in standard English?

Reviewer #1: Yes

5. Review Comments to the Author

Reviewer #1: The manuscript by Tatsuya Nishi, et al describes a series of animal experiments that were performed to evaluate the minimum infectious dose required to infect pigs with FMDV O/HKN/1/2015 of the CATAHY topotype, as well as determine whether cattle were susceptible to this specific virus strain. The manuscript is largely descriptive, presenting data on viral shedding dynamics and serological responses. There are some issues with the interpretation and description of the cattle experiments (see below), but this is otherwise a largely well-written and concise report.

Major comments

The most critical issue with the interpretation of the experimental findings relates to the statement that the amount of virus shed by the infected cattle was “apparently insufficient” to infect other animals.

Although, this can be speculated upon based upon low levels of virus shed by the infected cattle, it cannot be concluded that these animals were not capable of infecting other animals as transmission was not evaluated by contact trials.

Additionally, the cattle in the experiment were infected by direct injection of virus into the tongues. This experimental approach is highly artificial as it circumvents the natural routs of FMDV exposure, and does therefore not provide much information as to whether these animals would have been susceptible to this virus under more natural exposure conditions. It would have been more informative if susceptible cattle had been exposed to the infected pigs to evaluate whether the cattle were susceptible to infection or not. It is, however, understandable that such transmission experiments are highly resource demanding and may therefore not have been possible to perform. But, the interpretation of the findings regarding the infection of cattle needs to be adjusted to reflect what can actually be concluded based on the available data.

Additionally, viral titers in clinical samples are in reported as TCID50/0.1ml. This is atypical as the conventional way of expressing viral titers would be TCID50/ml. It is also inconsistent and unclear within the paper as the viral doses used to infect the animals are expressed per ml. The viral titers that are expressed per 0.1ml need to be transformed into TCID50/ml. Specifically, reporting viral titers in oral fluids per ml would make it possible to compare those values to the viral quantities that were required to infect animals.

Minor comments

• Row 20: Change “infectious dose” to “minimum infectious dose”

• Row 42: I assume the authors are referring to aerosolized virus expelled in pigs’ breath (“their discharge”). Please edit the senetence for clarity

• Row 66: “In addition, to our knowledge, only one report to date has described the susceptibility of cows to the porcinophilic strain (9).” In addition to reference #9, the authors have in other places cited Pacheco et al doi: 10.1016/j.virol.2013.08.003. (reference #12), which also describes the susceptibility of cows to a similar deletion mutant. The same US group has also published additional works on the same subject.

• Row 70: “we used an intraoral infectious dose to investigate…” Should this say intraoral inoculation?

• Rows 77-79. Please re-write this sentence. In current version, it is not clear whether the pigs received the exact doses reported, or 10^1.5-fold dilutions thereof (I am guessing the former, but it is unclear)

• Row 86: Change “sub-hoof” to accessory digit or dew claw

6. PLOS authors have the option to publish the peer review history of their article (what does this mean?). If published, this will include your full peer review and any attached files.

Reviewer #1: No

---

## [Author Response · Author response to Decision Letter 0]

9 Dec 2020

Thank you for reviewing our manuscript PONE-D-20-32103 entitled “Quantitative analysis of infection dynamics of foot-and-mouth disease virus strain O/CATHAY in pigs and cattle”. According to your suggestion, the manuscript was revised as follows. I hope this revised manuscript will be suitable for publication in the PLOS ONE.

Responses to Editor

Comment 1) Please ensure that your manuscript meets PLOS ONE's style requirements, including those for file naming.

 Answer. According to your pointing out, I have revised authors’ affiliations, whole main body of the manuscript, and file naming to meet the style requirements.

Comment 2) We note that you have included the phrase “data not shown” in your manuscript. Unfortunately, this does not meet our data sharing requirements. PLOS does not permit references to inaccessible data. We require that authors provide all relevant data within the paper, Supporting Information files, or in an acceptable, public repository. Please add a citation to support this phrase or upload the data that corresponds with these findings to a stable repository (such as Figshare or Dryad) and provide and URLs, DOIs, or accession numbers that may be used to access these data. Or, if the data are not a core part of the research being presented in your study, we ask that you remove the phrase that refers to these data.

 Answer. According to your comment, all relevant data were provided in the revised manuscript. The data of antibody detection using a solid phase competitive ELISA kit (SPCE) were provided in Table 3 (lines 143-148, 164-165 in the revised manuscript). The whole genome sequence of O/HKN/1/2015 have been deposited in GenBank under accession number LC595604. The nucleotide sequences of O/HKN/1/2015 obtained from sera from cows were also provided as a supplemental file (lines 185, 191-192, 358-359, S1 File in the revised manuscript).

Comment 3) In your Methods section, please provide additional information on the animal research and ensure you have included details on whether animals died before the end of virus infection experiment; if so, specify the number of animals found dead and the reasons for the death.

 Answer. In this study, all pigs and cattle survived until the end of the experimental period. I added this information at lines 92 in the revised manuscript.

 

Responses to Reviewer

Comment 1) The most critical issue with the interpretation of the experimental findings relates to the statement that the amount of virus shed by the infected cattle was “apparently insufficient” to infect other animals. Although, this can be speculated upon based upon low levels of virus shed by the infected cattle, it cannot be concluded that these animals were not capable of infecting other animals as transmission was not evaluated by contact trials. 

Additionally, the cattle in the experiment were infected by direct injection of virus into the tongues. This experimental approach is highly artificial as it circumvents the natural routs of FMDV exposure, and does therefore not provide much information as to whether these animals would have been susceptible to this virus under more natural exposure conditions. It would have been more informative if susceptible cattle had been exposed to the infected pigs to evaluate whether the cattle were susceptible to infection or not. It is, however, understandable that such transmission experiments are highly resource demanding and may therefore not have been possible to perform. But, the interpretation of the findings regarding the infection of cattle needs to be adjusted to reflect what can actually be concluded based on the available data.

 Answer. Just as your comment, the greatest weakness of our study is lacking data of contact trials and thus, cannot conclude possibility of infecting other animals as transmission and susceptibility in cattle. In the present study, the cattle were artificially infected with porcinophilic strain of FMDV by subepidermo-lingual inoculation. Analyses of virus infectivity titers, FMDV-gene and antibody detection from their clinical samples were performed. The nucleotide sequences of viruses obtained from infected the cattle were also confirmed. We expect these data could indicate the dynamics of the porcinophilic virus in cattle, possibility of propagation, adaptation or persistence, and the time-course of antibody response. We believe these findings could contribute appropriate diagnosis and surveillance of this topotype of virus. According to your comment, interpretations of the findings were adjusted in the revised manuscript (lines 26-27, 64-66, 216-218, 221-225, 230-232).

Comment 2) Additionally, viral titers in clinical samples are in reported as TCID50/0.1ml. This is atypical as the conventional way of expressing viral titers would be TCID50/ml. It is also inconsistent and unclear within the paper as the viral doses used to infect the animals are expressed per ml. The viral titers that are expressed per 0.1ml need to be transformed into TCID50/ml. Specifically, reporting viral titers in oral fluids per ml would make it possible to compare those values to the viral quantities that were required to infect animals.

 Answer. According to your advice, viral titers in clinical samples and the inoculum used to infect the animals were expressed as TCID50/ml in the whole revised manuscript (lines 75, 80, 130, 155, 164, 204, 222). 

Comment 3) Change “infectious dose” to “minimum infectious dose”

 Answer. According to your comment, the words were changed in the revised manuscript (line 19).

Comment 4) I assume the authors are referring to aerosolized virus expelled in pigs’ breath (“their discharge”). Please edit the senetence for clarity.

 Answer. According to your comment, the sentence were edited in the revised manuscript (line 41).

Comment 5) “In addition, to our knowledge, only one report to date has described the susceptibility of cows to the porcinophilic strain (9).” In addition to reference #9, the authors have in other places cited Pacheco et al doi: 10.1016/j.virol.2013.08.003. (reference #12), which also describes the susceptibility of cows to a similar deletion mutant. The same US group has also published additional works on the same subject.

 Answer. As mentioned above, the data of the present study does not demonstrate susceptibility of cows to the porcinophilic strain as your comment. The sentence was fully modified in the revised manuscript (lines 64-66).

Comment 6) “we used an intraoral infectious dose to investigate” Should this say intraoral inoculation?

 Answer. According to your comment, the sentence was removed in the revised manuscript (line 68).

Comment 7) Please re-write this sentence. In current version, it is not clear whether the pigs received the exact doses reported, or 10^1.5-fold dilutions thereof (I am guessing the former, but it is unclear)

 Answer. According to your comment, the sentence was modified to indicate the pigs received the exact doses in the revised manuscript (line 74).

Comment 8) Change “sub-hoof” to accessory digit or dew claw.

 Answer. According to your comment, the word was changed in the revised manuscript (line 83).

---

## [Editor Report · Decision Letter 1]

8 Jan 2021

Quantitative analysis of infection dynamics of foot-and-mouth disease virus strain O/CATHAY in pigs and cattle

PONE-D-20-32103R1

Dear Dr. Fukai,

We’re pleased to inform you that your manuscript has been judged scientifically suitable for publication and will be formally accepted for publication once it meets all outstanding technical requirements.

Kind regards,

Douglas Gladue, Ph.D

Academic Editor

PLOS ONE
---

## [Editor Report · Acceptance letter]

12 Jan 2021

PONE-D-20-32103R1 

Quantitative analysis of infection dynamics of foot-and-mouth disease virus strain O/CATHAY in pigs and cattle 

Dear Dr. Fukai:

I'm pleased to inform you that your manuscript has been deemed suitable for publication in PLOS ONE. Congratulations! Your manuscript is now with our production department. 

Kind regards, 

on behalf of

Dr. Douglas Gladue 

Academic Editor

PLOS ONE